# Probing the Drug Dynamics of Chemotherapeutics Using Metasurface-Enhanced Infrared Reflection Spectroscopy of Live Cells

**DOI:** 10.3390/cells11101600

**Published:** 2022-05-10

**Authors:** Po-Ting Shen, Steven H. Huang, Zhouyang Huang, Justin J. Wilson, Gennady Shvets

**Affiliations:** 1School of Applied and Engineering Physics, Cornell University, Ithaca, NY 14853, USA; ps944@cornell.edu (P.-T.S.); hh623@cornell.edu (S.H.H.); 2Department of Chemistry and Chemical Biology, Cornell University, Ithaca, NY 14853, USA; zh265@cornell.edu (Z.H.); jjw275@cornell.edu (J.J.W.)

**Keywords:** FTIR, metasurface, biosensing, label-free, in vitro, cancer cells, drug dynamics, metal complexes, cancer drug discovery

## Abstract

Infrared spectroscopy has drawn considerable interest in biological applications, but the measurement of live cells is impeded by the attenuation of infrared light in water. Metasurface-enhanced infrared reflection spectroscopy (MEIRS) had been shown to mitigate the problem, enhance the cellular infrared signal through surface-enhanced infrared absorption, and encode the cellular vibrational signatures in the reflectance spectrum at the same time. In this study, we used MEIRS to study the dynamic response of live cancer cells to a newly developed chemotherapeutic metal complex with distinct modes of action (MoAs): tricarbonyl rhenium isonitrile polypyridyl (TRIP). MEIRS measurements demonstrated that administering TRIP resulted in long-term (several hours) reduction in protein, lipid, and overall refractive index signals, and in short-term (tens of minutes) increase in these signals, consistent with the induction of endoplasmic reticulum stress. The unique tricarbonyl IR signature of TRIP in the bioorthogonal spectral window was monitored in real time, and was used as an infrared tag to detect the precise drug delivery time that was shown to be closely correlated with the onset of the phenotypic response. These results demonstrate that MEIRS is an effective label-free real-time cellular assay capable of detecting and interpreting the early phenotypic responses of cells to IR-tagged chemotherapeutics.

## 1. Introduction

Infrared (IR) spectroscopy is a non-destructive, label-free technique that provides molecular fingerprints through their vibrational modes. Recently, IR spectroscopy has attracted interest as a tool for quantitative analysis of biological samples. This interest is driven by the ease of sample preparation, as well as the rich spectrochemical information it provides [1]. Among the different applications, IR spectroscopy has demonstrated to be an effective tool for diagnosis and screening of viral infection, cancers, and other diseases based on blood serum samples [2,3,4]. IR spectroscopy in combination with chemometric and multivariate techniques has been demonstrated as a powerful analytic tool for cytological and histological diagnosis, for instance, in distinguishing between cancerous and healthy tissues [5,6,7,8,9,10]. In particular, the use of mid-IR spectrochemical imaging as a digital staining tool for stain-free histopathology has been an exciting development in recent years [11,12]. IR spectroscopy has also been applied to the investigation of the actions of various stimuli on cells and the resulting cellular responses, such as detecting immune cell activation, bacterial response to various antibiotics, and the modes of action (MoAs) of different families of drugs to the targeted cell lines [13,14,15,16,17].

While IR spectroscopy is a promising tool for biological analysis, conventional measurements involve end-time measurements on fixed and dried samples because it is challenging to measure live cells due to the strong attenuation of IR light in water. To probe live cells in water, either the sample needs to be probed in attenuated total reflection (ATR) configuration [18], or a very thin flow cell on the order of 10 μm needs to be used [19,20]. In its present form, ATR-FTIR is unsuitable as a tool for large scale cellular assays due to difficulty in integrating the internal reflection element with planar microplate format. The limited signal-to-noise ratio (SNR) of ATR-FTIR setups is also a concern, often necessitating a multi-bounce ATR element or averaging over many spectra [21]. High brightness sources, such as synchrotron radiation, supercontinuum, or quantum cascade laser (QCL), are available to improve the SNR and are particularly important for spectrochemical imaging applications [22,23,24,25]. However, synchrotron and supercontinuum sources are often limited by the availability of the instrument in a typical laboratory setting. Despite the recent advancement in multi-module QCL systems that extend the spectral coverage to much of the finger printing and amid I/II regions, in order to cover such a broad spectral window, the sweeping of the laser frequency and switching between several laser modules are necessary. Such frequency tuning makes QCL systems slow and more suitable for IR imaging at a few discrete frequencies.

Recently, our group has developed a metasurface-enhanced infrared reflection spectroscopy (MEIRS) approach to overcome this challenge [26,27,28]. In MEIRS, cells in proximity to an array of engineered plasmonic (typically gold) nanoantennas, known as the metasurface, are sensed through the evanescent fields at the hot spots of the nanoantennas. The IR absorption signal is encoded in the reflectance spectrum of the metasurface and, thus, the metasurface-attached cells can be probed without the IR light passing through water. In addition, MEIRS enhances the IR signals through surface-enhanced infrared absorption (SEIRA), which has been extensively studied for the measurement of biomolecules, such as supported lipid bilayers and protein monolayers [29,30,31,32]. Compared with ATR-FTIR that has field penetration around 1 μm [18], MEIRS provides a signal enhancement through SEIRA with smaller field penetration of roughly 100 nm [26,28]. The thickness of an adherent cell is on the order of a few micrometers, so the metasurface is selectively sensitive to the dynamic processes near the cell membrane on the basal side of the cells, such as cell adhesion, changes in cell membrane, and associated cytoskeleton reorganization. In addition, the metasurfaces are fabricated on planar substrates, which are compatible with microfluidics and microplate geometry. They can be scaled to modern high-throughput systems: a clear advantage over ATR-FTIR, where a large internal reflection element—either in its single- or multi-bounce formats—makes such integration difficult.

Previously, we applied MEIRS to the measurement of fixed and dried cells [26], cells captured through dielectrophoresis [27], as well as living cells grown directly on the metasurface [28]. In particular, we monitored the real-time spectroscopic changes as the cells were treated with low-concentration trypsin that induced intra-cellular signalling and detached the cells from metasurface, or with methyl-β-cyclodextrin that depleted the cholesterol in cellular membrane. We demonstrated that MEIRS spectra can be used to identify changes in protein, lipid, or refractive index signals in the proximity to the metasurface. Each signal has a different temporal behavior: an important feature that can be used to predict the MoA of various compounds [28]. The open question is whether MEIRS can be used to detect and characterize cellular responses to clinically relevant chemotherapeutics. The potential of MEIRS to detect cellular intake of chemical agents labeled by bioorthogonal [25,33] vibrational probes—chemical moieties whose spectroscopic signatures do not spectrally overlap with those of the endogenous cell molecules—also remains unexplored, even though the application of such probes to Raman and infrared spectroscopies are well-studied.

Here, we apply MEIRS to monitor the intracellular action of a newly developed metal-based anti-cancer agent: tricarbonyl rhenium isonitrile polypyridyl (TRIP) [34,35]. TRIP exhibits cytotoxic activity in multiple cancer cell lines by the induction of endoplasmic reticulum (ER) stress and unfolded protein response (UPR) [34,36], and eventually makes the cells lose viability. Although TRIP’s in vitro and in vivo activities, and MoA, have been studied previously, its kinetics is yet to be investigated. We sought to use MEIRS to monitor the cellular uptake of TRIP and the dynamics of cellular response upon TRIP treatment because TRIP has three non-degenerate tricarbonyl (C≡O) IR-active stretching modes in the bioorthogonal spectral range around 1900–2100 cm${}^{-1}$. We chose the human epidermoid carcinoma A431 cells as our model cell line. Cells were seeded on the metasurface and the metasurface was integrated with a microfluidic flow chamber with constant perfusion to sustain the cells over ≈10 h. TRIP was injected into the flow chamber while the IR spectra were collected to monitor the cells with Δt=1-min temporal resolution. The time-resolved spectra from MEIRS were analyzed in four judiciously chosen spectral windows—three biologically-important ones (corresponding to lipid and protein vibrations, as well as the overall refractive index shift of the plasmonic resonance) and one bioorthogonal—to monitor the in vitro drug response of cells to TRIP.

## 2. Materials and Methods

### 2.1. Cell Culture

Human epidermoid carcinoma cell line A431 (acquired from the American Type Culture Collection) was used for this work. Cells with passage number <10 were used for all experiments. Cells were cultured in a standard cell incubator with 5% CO${}_{2}$ and 37 ${}^{\circ }C$ in Dulbecco’s Modified Eagle Medium (DMEM, Gibco${}^{\mathrm{TM}}$, Thermo Fisher Scientific, Waltham, MA, USA) supplemented with 10% fetal bovine serum (FBS, Gibco${}^{\mathrm{TM}}$, Thermo Fisher Scientific) and 1% penicillin/streptomycin (Gibco${}^{\mathrm{TM}}$, Thermo Fisher Scientific).

Prior to seeding them on the metasurface (shown in Figure 1a), cells were dissociated from the culture flask using 0.25% trypsin-EDTA (trypsin-EDTA 0.25% with phenol red, Gibco${}^{\mathrm{TM}}$, Thermo Fisher Scientific). The metasurface sample was placed in a 12-well plate and cells were seeded in DMEM at approximately 200,000 cells per mL. The cells were allowed to proliferate to reach confluent coverage on the metasurface before measurements.

### 2.2. Viability Assay

The in vitro anti-cancer activity of TRIP was evaluated in A431 cells by 3-(4,5- dimethylthiazol-2-yl)-2,5-diphenyltetrazolium bromide (MTT) assay (shown in Figure 2). A431 cells were seeded in 96-well plates with 8000 cells/well in 100 μL of growth media and allowed to reattach overnight. The following day, the culture media was removed and replaced with 200 μL of media containing varying concentrations of the complex. The cells were incubated for 6 h. The drug-containing media was then removed and replaced with fresh culture media without drugs. The cells were incubated for an additional 42 h to ensure that the cells were in the logarithmic growth phase and that the cells had adequate time to regrow after exposure to the complexes. The media was removed from the wells, and MTT in DMEM (200 μL, 1 mg/mL) was added. After 4 h, the media was removed, and the purple formazan crystals were dissolved in 200 μL of an 8:1 mixture of dimethyl sulfoxide (DMSO) and pH = 10 glycine buffer. The absorbance at 570 nm in each well was measured using a microplate reader (Synergy HT, BioTek, Winooski, VT, USA). Cell viability was determined by normalizing the absorbance of the treated wells to untreated wells. Results were reported as the average cell viability of six replicates per concentration from three independent trials. To prevent acute poisoning to the cells, and to ascertain the sensitivity of the MEIRS assay, we selected the TRIP concentration, $C^{\mathrm{TRIP}}=5.0$
μM, below the half maximal inhibitory concentration (IC${}_{50}$) for all our spectroscopic measurements.

### 2.3. Metasurface Fabrication

Gold plasmonic metasurfaces were fabricated on 12.5 mm × 12.5 mm × 0.5 mm (width × length × thickness) CaF${}_{2}$ substrates. Electron beam lithography was used to pattern the metasurface with an e-beam lithography system (JEOL9500, JEOL Ltd., Akishima-shi, Japan). The patterns were written on a spin-coated 200-nm-thick poly(methyl methacrylate) (PMMA) polymer layer on the substrate with 5 nA current. Then, the patterns were developed in methyl isobutyl ketone (MIBK): isopropanol solution at the ratio of 1:3 for 90 s. Gold was deposited using electron beam physical vapor deposition. A total of 5 nm of chromium was deposited for adhesion and 70 nm of 99.999% pure gold was later deposited as the main material of the meta-atoms. The dimension of a pixel of the metasurface arrays was 300 μm × 300 μm (width × length). Each meta-atom’s unit cell dimension is approximately 1.458
μm × 1.701
μm (width × length), i.e., a metasurface pixel comprises a 205×176 array of π-shaped meta-atoms. Additional details of the metasurface geometry are described elsewhere [28]. Figure 1a left is the scanning electron microscope (SEM) image of fixed A431 cells on the meta-atoms of the metasurface. The left corner is a zoomed-in meta-atom image. Figure 1a right is the phase contrast microscopy image of live A431 cells on a single metasurface pixel.

### 2.4. FTIR Spectroscopy

A schematic illustration of the measurement setup, described in details elsewhere [28], is shown in Figure 1b. The cell-covered metasurface on the CaF${}_{2}$ substrate was mechanically clamped to a polydimethylsiloxane (PDMS) microfluidic chamber. The PDMS chamber was perfused with L-15 cell culture medium (Leibovitz’s L-15 Medium, Gibco${}^{\mathrm{TM}}$, Thermo Fisher Scientific, designed to maintain physiological pH at ambient air condition) supplemented with 1% antibiotic-antimycotic (Gibco${}^{\mathrm{TM}}$, Thermo Fisher Scientific) throughout the measurement. Constant perfusion was maintained at 0.1
μL/s using a programmable syringe pump system (Nemesys Low Pressure Syringe Pump, CETONI GmbH, Korbuβen, Germany). A microscope stage heater was used to heat the perfusion medium and the flow chamber to 37 ${}^{\circ }$C. The cells in the chamber were placed upside down and sat for an hour to adapt to the new environment before any measurement. For TRIP treatment, TRIP was dissolved in L-15 medium to make 5.0 μM TRIP-L-15 solution, and injected into the flow chamber at 0.1
μL/s using the same syringe pump system while the spectra were collected.

An FTIR spectrometer (VERTEX 70, Bruker, Billerica, MA, USA) coupled with an IR microscope (HYPERION 3000, Bruker) with a liquid-nitrogen cooled mercury cadmium telluride (MCT) detector was used to collect the IR spectra with Δω=4 cm${}^{-1}$ spectral and Δt=1 min temporal resolutions. To increase the SNR, the spectra were averaged over 120 scans. The spectrometer and microscope were purged with dry air. IR spectra were collected in the reflectance mode from the metasurface through the CaF${}_{2}$ substrate, with the attached cells facing down. In the reflectance mode, the background was measured on a patch of 300 μm × 300 μm film of gold (70 nm thick), fabricated on the same chip with the metasurface and measured through the 0.5 mm CaF${}_{2}$ substrate, prior to sample measurement. We note that cells adhered on this gold patch as well, but the 70 nm gold film was completely reflective in the measured wavelength range. The gold patch merely acted as a gold mirror for the purpose of the background measurement. The microscope aperture was set to a 200 μm × 200 μm square area. Mertz phase correction and the three-term Blackman–Harris apodization function were used. Water vapor correction was applied to the collected spectra using the built-in atmospheric compensation function in Bruker OPUS software. Spectra between 2262 and 2412 cm${}^{-1}$ were excluded to avoid the contribution from the strong CO${}_{2}$ absorption lines in that range. The reflectance spectrum of the metasurface with live cells in culture medium (without drug treatment), R(ω,t=0), is shown in Figure 1c. The vibrational modes of the cells, as well as that of water, can be seen overlapped on top of the metasurface’s broad plasmonic resonance. The absorbance spectrum A(ω,t=0) from the untreated cells on the metasurface shown in Figure 1d, as well as the time-dependent differential absorbance ΔA(ω,t), are defined as follows:(1)$$A\left(\omega ,t\right)=-\log_{10}\left[R\left(\omega ,t\right)/R_{0}\left(\omega \right)\right],\;\;\;\;\Delta A\left(\omega ,t\right)=A\left(\omega ,t\right)-A\left(\omega ,t=0\right),$$
where R(ω,t) is the time-dependent reflectance spectrum of the metasurface with cells and $R_{0}\left(\omega \right)$ is the reflectance spectrum of the metasurface with no cells remaining (i.e., removed by trypsinization). Different vibrational modes attributed to the endogenous biomolecules in cells can be clearly seen here. While we refer to the quantity A(ω,t) as absorbance, we note that it is not identical to the traditionally defined absorbance of IR light passing through a monolayer of cells on a flat surface. For example, because of the resonant (i.e., frequency-dependent) nature of the metasurface response, the two amide peaks may have unconventional relative magnitudes. Despite its limitations, the frequency dependence of A(ω,t=0) shown in Figure 1d has a more intuitive appeal than that of R(ω,t=0) plotted in Figure 1c.

### 2.5. Spectral Data Analysis

For each drug treatment measurement, we have a large number of absorbance spectra collected over time as the cells respond to TRIP. Although the two-dimensional data encoded in A(ω,t) is information-rich, it can be difficult to interpret. To assist with interpretation, we extracted the time-dependent IR vibrational signal attributed to a different class of molecules and refractive index change. We focused on certain spectral windows for protein, lipids, plasmonic resonance shift, as well as TRIP’s C≡O stretching, and performed least square linear regression in each spectral window using a set of reference spectra. For the signals of cellular origin (i.e., proteins, lipids, as well as plasmonic resonance shift), we plotted the absorption spectrum of the total cell signal A(ω,t=0) in Figure 1d as the cell response reference (CRR) spectrum. Specifically, for the protein and lipid differential absorbance spectra, we performed the linear regression using the second derivatives of the differential absorption spectra (i.e., $\partial^{2}A\left(\omega ,t\right)/\partial \omega^{2}$) and of the CRR (i.e., $\partial^{2}A\left(\omega ,t=0\right)/\partial \omega^{2}$). For the plasmonic resonance shift, the linear regression was performed using undifferentiated spectra, i.e., ΔA(ω,t) and A(ω,t=0), respectively. For the analysis of the time-dependent TRIP’s IR tag signal originating from the C≡O stretching, we first calculated the tricarbonyl absorbance spectrum on the metasurface without cells, and then ran the linear regression on its second derivative with respect to ω.

Formally, the regression relationship between the differential absorption spectra $\Delta A^{(n)\;win}\left(\omega ,t\right)$ (and its derivatives), the time-dependent IR vibration signal $S^{win}\left(t\right)$, and the CRR spectra $L^{(n)\;win}\left(\omega \right)$ (and its corresponding derivatives) can be written as:(2)$$\Delta A^{(n)\;win}\left(\omega ,t\right)=S^{win}\left(t\right)L^{(n)\;win}\left(\omega \right)+E,$$
where $S^{win}\left(t\right)$ were obtained by the least-squares linear regression and *E* is the minimized error term. Here, n= 0 or 2 denotes *n*-th derivative of the absorbance spectrum (n=0 means no derivative is taken). win denotes the different spectral windows: plasmonic resonance (superscript “pl”): 1845<ω<2231 cm${}^{-1}$, proteins (superscript “prot”): 1498<ω<1807 cm${}^{-1}$, lipids (superscript “lip”): 2756<ω<3064 cm${}^{-1}$, and TRIP tricarbonyl (superscript “C≡O”): 1845<ω<2135 cm${}^{-1}$. The smoothing and differentiation parameters for the spectra are as follows: 21-point Savitzky–Golay (SG) smooth filter for the plasmonic resonance shift signal, 21-point SG differentiation for the protein and lipid signal, and 11-point SG differentiation for the TRIP’s C≡O stretching signal. The extracted time-dependent IR vibration signal $S^{win}\left(t\right)$ are presented in Figure 4a–d. Five replicates were done for each condition. The peak assignment of the IR vibration modes can be found in Table 1.

Note that this method of analyzing vibrational modes with second derivative and linear regression may seem overly complex compared to the popular method of baseline subtraction and peak integration. However, in MEIRS, the cellular vibrational signal overlaps in a complex manner with that from the plasmonic metasurface resonance, and we found the baseline subtraction and peak integration to be inadequate for separating the spectral changes due to plasmonic resonance shift and that due to vibrational modes. In particular, the TRIP’s C≡O stretching significantly overlaps with the plasmonic resonance shift of the metasurface. The second derivative method, with the appropriate smoothing window, takes advantage of the different bandwidth of these two spectral features; it was found to be effective in separating the two signals.

For different replicates of the drug treatment, we observed measurable delays in the arrival of TRIP as measured by MEIRS (presented in Figure 3). Originally, we expected the drug should arrive at the chamber in 50 min at a flow rate of 0.1
μL/s, which was confirmed by injecting a dye solution to our perfusion system. However, TRIP arrived later than 70 min. We have confirmed that the delay in the timing of TRIP detection is present even for the metasurface without cells. This suggests that the delay seen in the TRIP signal is not related to any cellular response, but rather likely due to some of the specific features of our experimental setup, such as: the need for constant low-speed flow of the cell culture to avoid exposing the cells to flow-induced shear stress, and relatively long tubing that can facilitate TRIP molecule adhesion to its interior when the flow speed is low. When cells are present on metasurfaces, this delay varies among the replicates. An example of the IR-tag signal $S^{C\equiv O}\left(t\right)$ from two (out of the five) replicates is shown in Figure 3a. For simplicity, we assume that the drug arrival time $t_{\mathrm{arr}}^{\left(j\right)}$ of the *j*’th replicate corresponds to the half-max of $S^{C\equiv O}\left(t\right)$. The two replicates presented in Figure 3a,b correspond to the earliest ($t_{\mathrm{arr}}^{\left(5\right)}\approx 85$ min) and latest ($t_{\mathrm{arr}}^{\left(4\right)}\approx 119$ min) arrival time, with the remaining three replicates arriving at the intermediate times $t_{\mathrm{arr}}^{\left(5\right)}<t_{\mathrm{arr}}^{\left(3),(2),(1\right)}<t_{\mathrm{arr}}^{\left(4\right)}$ (see Appendix A).

Crucially, we have observed that the occurrence times $t_{\mathrm{peak}}^{\left(j\right)}$ of the first peak of the protein signal $S^{\mathrm{prot}}\left(t\right)$ plotted in Figure 3b showed similar latencies relative to their respective drug arrival times. For example, $t_{\mathrm{peak}}^{\left(5\right)}\approx 89$ min and $t_{\mathrm{peak}}^{\left(4\right)}\approx 123$ min, i.e., both protein signal peaks are delayed by approximately $\Delta t^{\left(j\right)}\equiv t_{\mathrm{peak}}^{\left(j\right)}-t_{\mathrm{arr}}^{\left(j\right)}\approx 4$ min from the previously defined drug arrival time. The latency times $\Delta t^{\left(j\right)}$ are plotted in Figure 3c as a function of the drug arrival times $t_{\mathrm{arr}}^{\left(j\right)}$ for all five replicates. Therefore, the measured arrival time can be used as a valuable *time fiducial* to appropriately time-calibrate the three cell-related responses $S^{\mathrm{prot}}\left(t\right)$, $S^{\mathrm{lip}}\left(t\right)$, and $S^{\mathrm{pl}}\left(t\right)$. Specifically, the arrival times of all replicates were shifted to the common value of $t_{0}=70$ min (initial rise in TRIP signal of the earliest arriving replicate), corresponding to $t_{\mathrm{arr}}^{\left(5\right)}\approx 85$ min labeled as a red diamond in Figure 3c. A comparison between the time-calibrated and uncalibrated signals is plotted in Appendix A.

As a final remark for calculating the time-dependent TRIP’s IR tag signal, we found that there were metasurface fabrication-related sensitivity variation that caused the variation in signal amplitude. That is, there was an amplitudinal variation for the three C≡O stretching peaks when the measurements were done on different batches of metasurface devices. A simple sensitivity calibration was performed by regressing the TRIP measurement spectra to the basis obtained from the without-cell TRIP measurement on the same device.

## 3. Results

### 3.1. Delivery of the TRIP Anti-Cancer Complex to Cells Can Be Monitored in Time with MEIRS

First, we demonstrate that the presence of TRIP molecules in the flow chamber can be monitored spectroscopically. Such determination can be made because the TRIP molecule absorbs at $\omega_{1}^{C\equiv O}=1928$ cm${}^{-1}$, $\omega_{2}^{C\equiv O}=1967$ cm${}^{-1}$, and $\omega_{3}^{C\equiv O}=2031$ cm${}^{-1}$ vibrational frequencies corresponding to three non-degenerate C≡O stretching modes of the tricarbonyl group [35]. Because the frequencies of these vibrational modes are spectrally localized in the bioorthogonal part of the spectrum (i.e., far away from any major vibrational frequencies of the essential biomolecules or water), the arrival of the IR-tagged compounds (such as TRIP) to a metasurface can be easily detected by MEIRS.

Time-dependent IR-tag signal $S^{C\equiv O}\left(t\right)$ is presented in Figure 4a, where the reference $L^{(2)\;C\equiv O}\left(\omega \right)$ is shown in the inset. The time-dependent signals quantified the absorption of any chemotherapeutic that contains a C≡O stretching-based IR tag (TRIP, in this case). The control medium contained no C≡O stretching IR tag. Therefore, only the TRIP-treated A431 cells showed any increase in the TRIP absorption signals. As a reference, TRIP measurements were also made using metasurface without cells.

The TRIP absorption signal with cells was characterized by a sharp increase in IR absorption at $t_{0}=70$ min, followed by a much slower increase starting from t=200 min. On the other hand, the increase in the TRIP absorption signal for metasurface without cells at $t_{0}=70$ min was even steeper than the with-cell measurement, followed by a much steeper increase starting from t=200 min. Comparing between the with-cell and no-cell curves, the no-cell TRIP signal rose faster and reached a higher amplitude plateau. This suggests that the intracellular concentration of TRIP is lower than the concentration of TRIP in the extracellular environment.

### 3.2. Analysis of Spectroscopic Changes of Drugs-Influenced Cells in Bio-Relevant Spectral Regions

Next, we analyzed the biological responses of A431 cells to TRIP by observing the temporal dynamics of $S^{win}\left(t\right)$ for the three biologically-relevant spectral ranges that were used to characterize: (i) the overall refractive index around the metasurface related to the cells, (ii) absorption in the protein, and (iii) lipid spectral windows.

While plasmonic (Fano) resonance, manifested as a broad strong peak of the initial absorptivity A(ω,t=0) in Figure 1d around ω≈2000 cm${}^{-1}$, is supported by plasmonic nanoantennas alone, it has biological relevance by virtue of its dependence on the effective refractive index of the environment surrounding the antennas. Therefore, any changes of plasmonic resonance can be attributed to the phenotypic changes of cells attached to the metasurface. Such changes include: the increase or decrease of cells’ adhesion to the metasurface, cytoskeletal rearrangements, and protein translocation to/from the cellular membrane.

Observing cellular responses to chemical compounds by measuring spectral shifts of plasmonic (and, in general, photonic) resonances due to refractive index changes is known as dynamic mass distribution (DMR) in resonant waveguide grating (RWG) based optical real-time cellular assays (RTCA) [38,39,40,41]. These RTCAs have been used in pharmacological studies [42,43] for some time. When confined to a single spectral window containing plasmonic resonance (orange shaded area in Figure 1c), MEIRS provides very similar information to that obtained from DMR in optical RTCAs [28]. Therefore, the plasmonic spectral window is very convenient for benchmarking MEIRS results against more traditional optical RTCAs, including surface-plasmon resonance (SPR) [44,45,46] or RWG [38] based phenotypical assays.

**Plasmonic resonance shift:** the time-dependent signal $S^{\mathrm{pl}}\left(t\right)$ of the plasmonic resonance shifts in the TRIP-treated cells and control are presented in Figure 4b with the plasmonic CRR spectra $L^{(0)\;\mathrm{pl}}\left(\omega \right)$ (1845<ω<2231 cm${}^{-1}$) plotted in the inset. Because the refractive index of the cell is higher than that of the cell culture medium, any evolution of cellular adherence to the metasurface is manifested as time-dependent refractive index change [43]. The resulting spectral shift of the plasmonic resonance manifested as a broad peak (see inset) in the absorbance spectrum in the plasmonic resonance window.

By observing $S^{\mathrm{pl}}\left(t\right)$, we note that the time-dependent signal slightly increased over time for the control group. On the other hand, the TRIP-treated cells exhibited biphasic behavior: initial rapid increase (around $t_{0}=70$ min) followed by a gradual decrease (after t≈150 min) of its refractive index signal. A spike of $S^{\mathrm{pl}}\left(t\right)$ around $t=t_{0}$ is observed in the TRIP-treated cells, but not in the control.

**Protein IR vibration:** the time-dependent signal $S^{\mathrm{prot}}\left(t\right)$ obtained from protein IR vibrations is plotted in Figure 4c with the inset showing the CRR for the protein window. Vibrational signatures in this spectral window includes the Amide I at $\omega_{A,I}\approx 1650$ cm${}^{-1}$ and amide II at $\omega_{A,\mathrm{II}}\approx 1550$ cm${}^{-1}$ bands [37,47]. The increase (decrease) of the $S^{\mathrm{prot}}\left(t\right)$ signals corresponds to the increase (decrease) in protein IR absorption. We observe a gradual increase in the protein signal for the control group.

The TRIP-treated cells exhibited biphasic behavior: initial rapid increase (around $t_{0}=70$ min) followed by a gradual decrease of its protein signal. The first phase is coincident with the refractive index spike observed in the plasmonic resonance signals $S^{\mathrm{pl}}\left(t\right)$ discussed earlier.

**Lipid IR vibration:** the time-dependent signal $S^{\mathrm{lip}}\left(t\right)$ obtained from lipid IR vibrations is plotted in Figure 4d with the inset showing the CRR for the lipid window. Spectral signatures of the window include -CH${}_{2}$- symmetric stretching at 2862 cm${}^{-1}$, -CH${}_{2}$- antisymmetric stretching at 2925 cm${}^{-1}$, and -CH${}_{3}$ antisymmetric stretching at 2970 cm${}^{-1}$. These vibrations are typically assigned to the cellular lipid [37], and, in the particular case of MEIRS, to the lipids in the cellular membrane [28]. The increase (decrease) of the $S^{\mathrm{lip}}\left(t\right)$ signals corresponds to the increase (decrease) in lipid IR absorption.

We observe from Figure 4d that the TRIP-treated cells showed a significant decrease in the lipid signals, whereas the control group showed a slight increase over time. The SNR for the lipid signal was relatively low compared with the protein signal. This is attributed to the very thin (under 10 nm) cell membrane from where most of the signal originates, and the low infrared absorption cross section of -CH${}_{2}$ stretches compared with the amide bands.

## 4. Discussion

### 4.1. Interpretation of the Spectral Signals

The time-dependent signal for protein, lipids, as well as plasmonic resonance shift shared a similar trend. For the control group, all three of them showed a steady increase in the signal, which may be interpreted as increasing cellular adhesion or cellular growth. On the other hand, for TRIP-treated cells, a prominent feature in the time-dependent signals was a sharp increase in the protein and refractive index signals that coincided with the increase in C≡O stretching signal at $t_{0}=70$ min. Then, the signals quickly decreased again in about 30 min. Such behavior was seen in the lipid signals as well, but not very obvious due to the limited SNR. Following that, in the second phase, all three of the signals started to decrease at around t = 150 min.

The most likely origin of the initial peak after $t_{0}=70$ min is the protein translocation and cytoskeletal reorganization in response to Ca${}^{2+}$-related intracellular signaling [48]. One dominant mode of action of TRIP is ER stress [34,35], which can cause the release of Ca${}^{2+}$ ions from the ER. Ca${}^{2+}$ ions, on the other hand, is involved in the signal transduction of many pathways such as the G-protein coupled receptor (GPCR) or the epidermal growth factor receptor (EGFR) pathway [38,40,41]. We have previously observed similar peaks in MEIRS cellular signals when the cells were exposed to trypsin, a protease-activated receptor 2 (PAR2) agonist, and this had been attributed to the protein translocation and cytoskeletal reorganization in response to the PAR2 pathway [28,39]. Thus, we hypothesize that TRIP also triggers a downstream cellular response mediated by Ca${}^{2+}$ ions.

The second phase’s origin is likely from the cell shrinkage or partial detachment. From the previous studies, ER stress and UPR caused by TRIP induce autophagy and apoptosis [34,35], which can lead to cell shrinkage and detachment. With phase contrast microscopy, we have observed the appearance of cytoplasmic vacuoles, which may be autophagic vesicles, cell shrinkage, and partial detachment starting at t=560 min and it became obvious after t=1000 min of TRIP treatment (Figure 4e). Our previous study has also shown that a rapid inhibition of protein translation occurs as early as 2 h after TRIP treatment [34], which could be another explanation for the decrease in cellular signal in the second phase.

Though protein, lipid, and refractive index shift signals shared a similar trend, difference can be observed in the relative amplitude of these signals at different times. In particular, the protein signal appears to have a second peak at around t=140 min, which is not observed in the lipid and refractive index signal. The origin of this peak is not clear, but it may be related to the accumulation of misfolded proteins and protein aggregation induced by ER stress due to TRIP [34].

### 4.2. Rapid Measurement of Drug Action Using MEIRS

Comparisons of the TRIP-treated cells with the control cells are presented in Figure 5a to evaluate the sensitivity of the MEIRS assay. We plotted the final signal change (at $t_{\mathrm{final}}=450$ min) seen in the protein, lipid, and refractive index signal across the five replicates. We also calculated the *p*-value of the time-calibrated biological signals at $t_{\mathrm{final}}$ with Welch’s unequal variance *t*-test. The TRIP-treated cells show highly statistically significant difference (p≤0.001) in the protein, lipid, and plasmonic (refractive index) signals at the end of the measurement.

Moreover, we have calculated the statistical significance of the TRIP treatment observed by MEIRS assay at every minute to evaluate how soon the action of TRIP can be seen in the cells (Figure 5b,c). The earliest statistical significance for the TRIP treatment came at around t=140 min in the plasmonic window and t=220 min in the protein window. See Appendix A (a) (protein signals) and (b) (plasmonic signals) for the data used in *p*-value calculation at each minute.

The decreased cellular signal seen by MEIRS is consistent with the loss of cell viability as measured by the standard MTT assay. Whereas the MTT assay typically requires long incubation time for cell growth and MTT reduction, MEIRS assay measures the cellular signal directly during TRIP treatment and can detect the loss of cell viability due to TRIP much earlier. MEIRS is also simpler and less labor-intensive because it does not involve any reagents, labels, or dyes. In future studies, it will be interesting to compare MEIRS with other conventional reporter dye-based cellular assays in monitoring cell viability and different modes of cell death.

## 5. Conclusions

MEIRS, an IR spectroscopy technique based on plasmonic metasurface, was used to monitor the action of TRIP, a novel metal-based chemotherapeutic drug, on live A431 cells. Apoptosis and loss of cell viability were observed through the reduction in cellular signal as seen in protein IR vibrations, lipid IR vibrations, as well as local refractive index change. The bioorthogonal C≡O stretching signatures of TRIP was used to determine the precise drug delivery time and helped to interpret the biological signals in the protein, lipid, and refractive index. MEIRS provides an appealing complementary tool to the MTT assay for measuring the course of cytotoxicity in real time. In the future, we wish to scale-up MEIRS to the high-throughput systems, enabling simultaneous measurement of different drugs at different concentrations.

## Figures and Tables

**Figure 1 cells-11-01600-f001:**
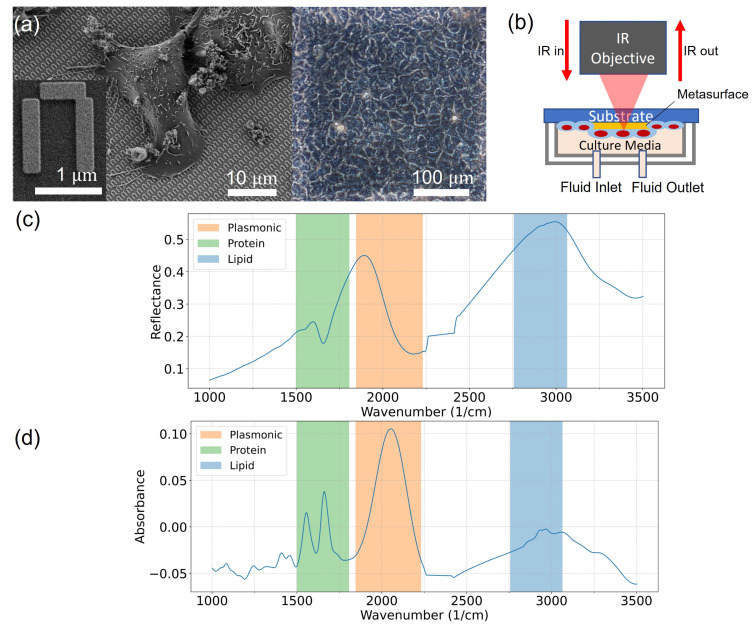
MEIRS setup and live cells spectra. (**a**) Left: SEM image of A431 cells cultured, subsequently fixed on the metasurface. The left corner is a zoomed-in single meta-atom. Right: phase contrast microscopy image of live A431 cells on a pixel of the metasurface (shaded square area of 300 μm × 300 μm). (**b**) Measurement setup: metasurface with live cells is sealed in a PDMS microfluidic chamber used for delivering cell culture and drugs. Reflectance spectra were collected from the top through IR-transparent substrate. The setup is described elsewhere [28]. (**c**) Reflectance spectrum R(ω) of A431 cells on the metasurface prior to drug treatment. Bio-relevant spectral windows are shaded: proteins (green, 1498<ω<1807 cm${}^{-1}$), plasmonic resonance (orange, 1845<ω<2231 cm${}^{-1}$), and lipids (blue, 2756<ω<3064 cm${}^{-1}$). Reflectance in the plasmonic resonance region is a proxy for refractive index changes due to cells’ presence on the metasurface. (**d**) Total cell absorbance spectrum A(ω) of A431 cells on the metasurface extracted from the reflectance spectrum R(ω) as $A\left(\omega \right)=-\log_{10}\left[R\left(\omega \right)/R_{0}\left(\omega \right)\right]$, where $R_{0}\left(\omega \right)$ is the average reflectance spectrum of the metasurface with no cells.

**Figure 2 cells-11-01600-f002:**
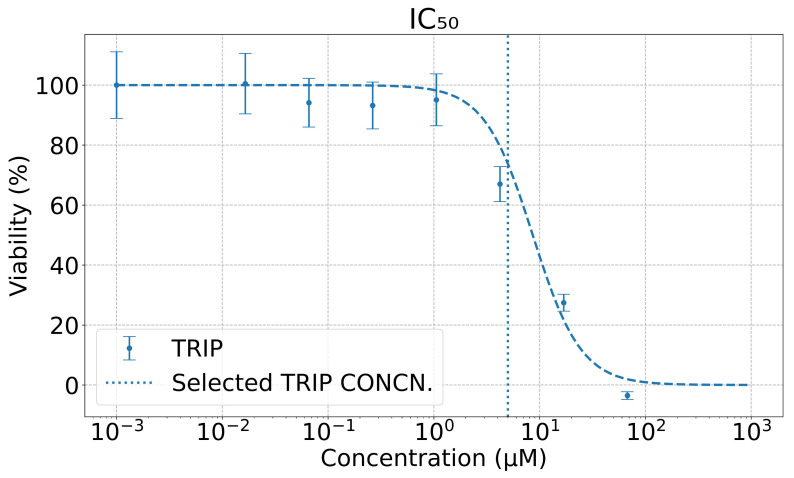
The dose-response curves of TRIP (blue line). The half maximal inhibitory concentrations (IC${}_{50}$) of the drug is ${\mathrm{IC}}_{50}^{\mathrm{TRIP}}=10.8\pm 2.8$
μM. A below-IC${}_{50}$ concentration, $C^{\mathrm{TRIP}}=5.0$
μM (vertical blue line), was used in all MEIRS assays.

**Figure 3 cells-11-01600-f003:**
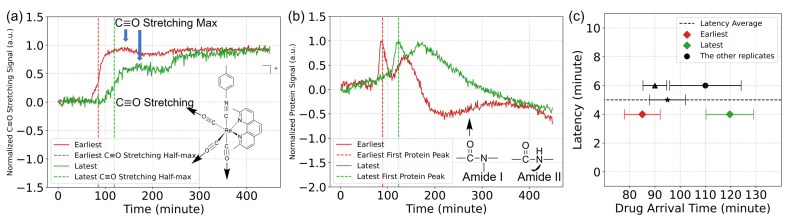
(**a**) Normalized C≡O stretch (IR-tag) signals for two representative measurements with different drug arrival times: the earliest (red line) and the latest (green line). Dashed vertical lines: arrival times $t_{\mathrm{arr}}$ of the drug (see text for definition). Inset: molecular structure of TRIP and the stretches of its C≡O moiety. (**b**) Same as (**a**), but for the corresponding protein signals $S^{\mathrm{prot}}\left(t\right)$. Inset: amide I and II vibrations. (**c**) Latency $\Delta t\equiv t_{\mathrm{peak}}-t_{\mathrm{arr}}$ vs arrival $t_{\mathrm{arr}}$ times for the five replicates.

**Figure 4 cells-11-01600-f004:**
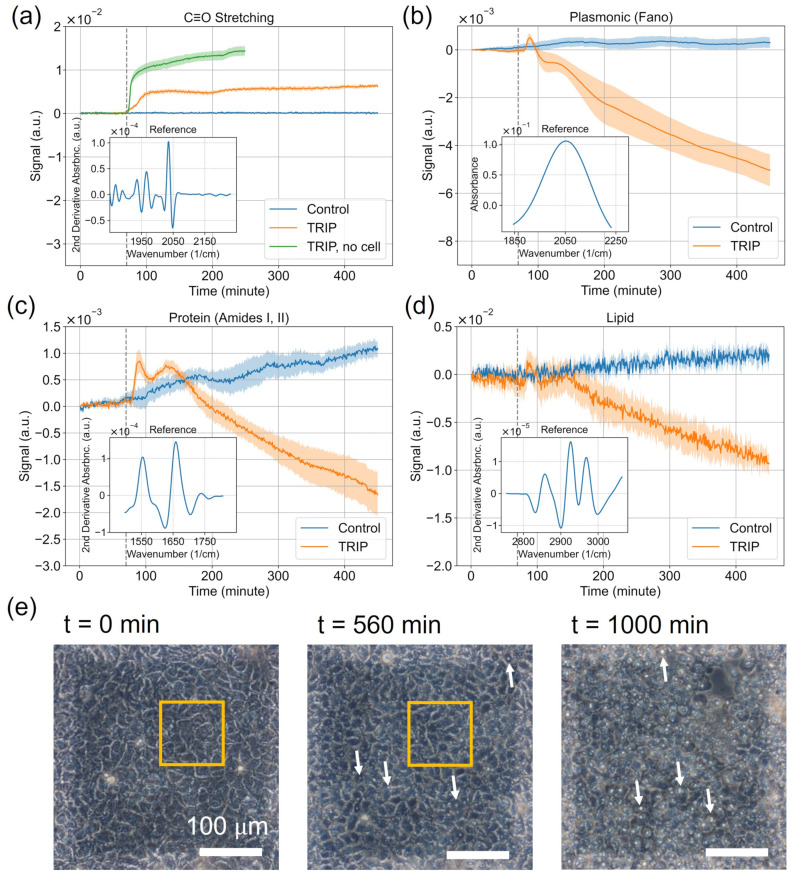
Time-dependent MEIRS signals of A431 cells under drug treatment with TRIP ($C_{\mathrm{TRIP}}=5.0$
μM) for the four spectral windows: bioorthogonal (**a**) and cell-related (**b**–**d**). Insets: reference spectra used for linear regression. Blue (orange) solid curves: control (drug-treated) cell responses S(t) representing the mean of five replicates. Shaded regions: standard errors. Green curve in (**a**): IR-tag signal of TRIP without cells. Drug arrival times are shifted to $t_{0}=70$ min (vertical dashed line). (**a**) $S^{C\equiv O}\left(t\right)$ for the intrinsic IR-tag signal of the TRIP compound. Inset: $L^{(2)\;\mathrm{IR}-\mathrm{tag}}\left(\omega \right)$, with peaks at $\omega_{1}^{C\equiv O}=1928$ cm${}^{-1}$, $\omega_{2}^{C\equiv O}=1967$ cm${}^{-1}$, and $\omega_{3}^{C\equiv O}=2031$ cm${}^{-1}$. (**b**) $S^{\mathrm{pl}}\left(t\right)$ for the plasmonic resonance shift signal. Inset: $L^{(0)\;\mathrm{pl}}\left(\omega \right)$ CRR spectrum for the plasmonic spectral window (1845<ω<2231 cm${}^{-1}$). (**c**) $S^{\mathrm{prot}}\left(t\right)$ for the protein signal. Inset: $L^{(2)\;\mathrm{prot}}\left(\omega \right)$ CRR spectrum for the protein spectral window (1498<ω<1807 cm${}^{-1}$). (**d**) $S^{\mathrm{lip}}\left(t\right)$ for the lipid signal. Inset: $L^{(2)\;\mathrm{lip}}\left(\omega \right)$ CRR spectrum for the lipids spectral window (2756<ω<3064 cm${}^{-1}$). Note that the signs of the second derivative reference spectra are flipped for more intuitive presentation. (**e**) The phase contrast microscopy images of A431 cells before, after t=560 min, and after t=1000 min TRIP treatment. White arrows: the cytoplasmic vacuoles as the signs of apoptosis. Yellow squares: marking of the same area to identify cell shrinkage after TRIP treatment. Shaded areas: $300\times 300\;{\mu m}^{2}$ metasurface.

**Figure 5 cells-11-01600-f005:**
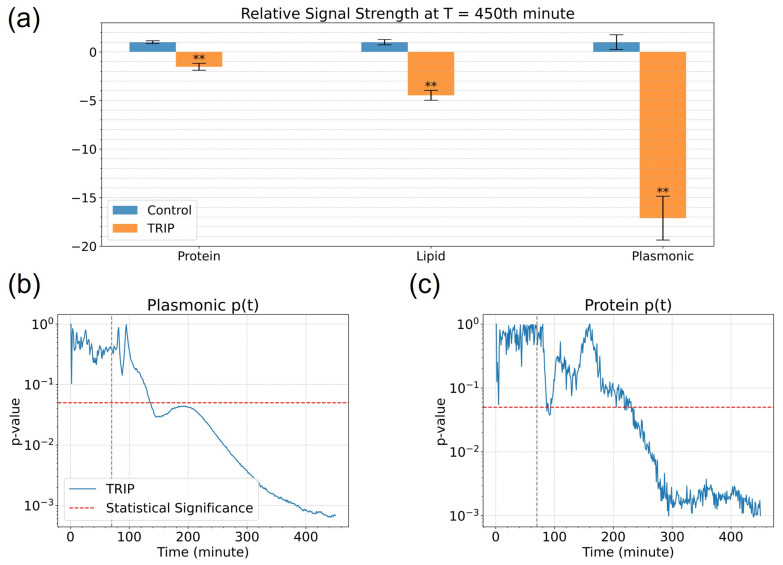
(**a**) Relative change in signal strength between t=0 min to t=450 min ($t_{\mathrm{final}}$) in three spectral windows, normalized to the control group. **: p≤0.001. The TRIP-treated ($C_{\mathrm{TRIP}}=5.0$
μM) group showed statistical significance in the protein, lipid, and plasmonic (i.e., refractive index) signals compared to the control. (**b**) *p*-values of the TRIP treatment at different times as seen through the MEIRS plasmonic resonance signal. The TRIP-treated cells showed the earliest statistically significant difference through *p*-value at around t=140 min. Initial drug arrival: $t_{0}=70$ min (vertical dashed line). Statistical significance: p=0.05 (horizontal red dashed line). (**c**) *p*-values of the TRIP treatment at different times as seen through the MEIRS protein vibration signal. The TRIP-treated cells showed the earliest statistically significant difference through *p*-value in the protein signals at around t=220 min.

**Table 1 cells-11-01600-t001:** The vibrational peak assignments for each spectral windows. δ: bending; *v*: stretching.

Origin	Protein [26]	Lipid [37]	Plasmonic [28]	TRIP [35]
			(Refractive Index)	
Spectral Window				
for	1498–1807	2756–3064	1845–2231	1845–2135
Reference Spectrum				(bioorthogonal)
(cm${}^{-1}$)				
	Amide II ≈ 1550:	-CH${}_{2}$- antisymm. stretching:		
	δ(NH) + *v*(CN)	2925		
Vibrational Modes	Amide I ≈ 1660:	-CH${}_{2}$- symm. stretching:	Fano resonance:	C≡O stretching:
(cm${}^{-1}$)	*v*(C=O) + δ(NH),	2862	2080	1928, 1967, 2031
	merged with	CH${}_{3}$ antisymm. stretching:		
	H${}_{2}$O absorption	2970		

## Data Availability

Data are contained within the article or Appendix A. Raw data are available upon request.

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
