# Peer review of "Probing the Drug Dynamics of Chemotherapeutics Using Metasurface-Enhanced Infrared Reflection Spectroscopy of Live Cells"

_cells, 2022, doi:10.3390/cells11101600_

Round 1

Reviewer 1 Report

The manuscript covers an interesting study of drug dynamics using a special reflection technique, i.e. metasurface-enhanced infrared reflection spectroscopy. The experimental design is fine and the spectral phenomena connected to the mode-of-action monitoring are clearly described. A few minor changes are required: 

The statement in the introduction that no commercial QCLs operating around 3000 cm-1exist is wrong. Quantum and Interband Cascade Lasers (QCLs and ICLs) are available with center wavelengths between 3 - 11 µm (cw and pulsed); there are DFB interband cascade lasers at any wavelength between 2.8 m and 6.5 µm.

Several references had the term “Fourier transform” which needs a capital first letter for Fourier. The adjective description “bio-orthogonal spectral range around 1900 2100 cm-1” has never occurred in my spectroscopist vocabulary.

The spectral windows mentioned in Table 1 should be reported with rounded off interval limits.

Author Response

The manuscript covers an interesting study of drug dynamics using a special reflection technique, i.e. metasurface-enhanced infrared reflection spectroscopy. The experimental design is fine and the spectral phenomena connected to the mode-of-action monitoring are clearly described. A few minor changes are required:

  1. The statement in the introduction that no commercial QCLs operating around 3000 cm-1exist is wrong. Quantum and Interband Cascade Lasers (QCLs and ICLs) are available with center wavelengths between 3 - 11 µm (cw and pulsed); there are DFB interband cascade lasers at any wavelength between 2.8 m and 6.5 µm.

The reviewer is right. There are ICLs that go to 3000cm-1. However, QCL and ICL need frequency tuning and switching modules to cover a wide spectral range. They slow the acquisition. FTIR-based techniques are better for broad spectral range.

Action: Remove the mentioned texts. Replace with “Despite the recent advancement in multi-module QCL systems that extend the spectral coverage to much of the finger printing and amid I/II regions, in order to cover such broad spectral window the sweeping of the laser frequency and switching between several laser modules are necessary. Such frequency tuning makes QCL systems slow and more suitable for IR imaging at a few discrete frequencies.”

  1. Several references had the term “Fourier transform” which needs a capital first letter for Fourier.

Action: The capitalization of “Fourier” is fixed now.

  1. The adjective description “bio-orthogonal spectral range around 1900 2100 cm-1” has never occurred in my spectroscopist vocabulary.

The phrase ”bio-orthogonal” has been used previously in literature (Ref. 24 and 32) to refer to IR probes with no spectral overlap with intrinsic biomolecules in the cells.

Action: We have moved the two references from the end of the paragraph to where “bio-orthogonal” first appeared.

  1. The spectral windows mentioned in Table 1 should be reported with rounded off interval limits.

The values for spectral window range in Table 1 are the exact numbers used in Section 2.5 for spectral analysis and we believe that it’s better to keep it consistent.

Action: None.

Reviewer 2 Report

The present manuscript presents valuable and interestingly information regarding the application of a method based on infra-red spectroscopy (MEIRS) to monitor live animal adherent cells along a defined period when submitted to a specific drug (TRIP).

In general, the manuscript is well structured, and data is well presented. However, I have some minor comments concerning data presentations, and doubts about the experimental setup and data interpretation. 

On the MEIR experiments, was TRIP added continuously, i.e., along the whole experiment?

To evaluate the hypothesis to explain the time delay, we should consider:

  • What is the theoretical time to TRIP reach the cells in the MEIRS setup, considering the medium flow rate and tube length?
  • Was this setup equal to setup used in reference nº 27? If so, when authors added for example trypsin was the signal received immediately after trypsin addition, or was also observed a delay, if so, what was that delay time?

On Fig. 2, is misleading to include in the same figure, the viability graph, and the microscopic images, since the viability assay was not conducted along MEIRS experiments. Authors did not measure viability of cells at T=0 and T=1000 min of the MEIRS experiments.

Just a comment: Would be interestingly to have run more replicated MEIRS experiments, to enable to stop some experiments at specific time intervals, to analyse the cells morphology, cells number and cells viability.

Line 363, be careful with interpretation since, at the beginning of the experiment, cells were already full confluent. When this happens, usually cells start to dye and not to grow. Furthermore, the cells division usually takes 20 hr.

To help interpreting, it is important to know if cells were putted upside down just before starting the MEIRS experiment? This is relevant for data interpretation since cells always need some adaptation period when putted in a new environment.

Also, protein denaturation leads to increase absorbance of amide bands. Consequently, absorbance increase could be from this phenomenon and not necessarily from protein new synthesis. For example, reference 33, indicates that TRIP starts by inducing unfolded protein that eventually leads to apoptosis. Therefore, the initial absorbance increase could be partly from protein unfolding, or even to increase protein expression, due to the cells response to the stress (TRIP).

Considering the hypotheses of “MOA”, authors observed “peaks” at defined periods, that are associated to cells macromolecular rearrangements/ or new synthesis. Authors did not prove with the present work that these rearrangements were due to Ca2+ or due to GPCR, or even at the process beginning (when absorbance increase) if were due to protein denaturation or protein synthesis.

So, I do not think that MOA was minimally proved. Consequently, I also do not agree with the manuscript title.

Line 390, please add a reference concerning TRIP inducing autophagy.

In introduction section, authors points in line 64 “basal membrane”, but I wonder if in this type of cultured cells there are a basal membrane. If so, I would request authors to include a reference.

All the Discussion section is a lithe confusing. Authors should review it.

Despite of all these comments, I think that the work is valuable. However, before publication all these doubts should be clarified, and the text corrected according to it.

Author Response

The present manuscript presents valuable and interestingly information regarding the application of a method based on infra-red spectroscopy (MEIRS) to monitor live animal adherent cells along a defined period when submitted to a specific drug (TRIP).

In general, the manuscript is well structured, and data is well presented. However, I have some minor comments concerning data presentations, and doubts about the experimental setup and data interpretation.

  1. On the MEIR experiments, was TRIP added continuously, i.e., along the whole experiment?

We did add TRIP along the whole experiment.

Action: It’s more clarified in the method: “For TRIP treatment, TRIP was dissolved in L-15 medium to make 5.0 μM TRIP-L-15 solution, and injected into the flow chamber at 0.1 μL/sec using the same syringe pump system while the spectra were collected.”

  1. To evaluate the hypothesis to explain the time delay, we should consider:

What is the theoretical time to TRIP reach the cells in the MEIRS setup, considering the medium flow rate and tube length?

Was this setup equal to setup used in reference nº 27? If so, when authors added for example trypsin was the signal received immediately after trypsin addition, or was also observed a delay, if so, what was that delay time?

The drug should be arriving at 45-50min at this flow rate. We also see immediate cell response from trypsin-related signaling without any delay around this time. So far, we have only seen such delay with TRIP.

Action: Detailed texts are added to describe the delay only occur with TRIP. “For different replicates of the drug treatment, we have observed measurable delays in the arrival of TRIP as measured by MEIRS (presented in Figure 3). Originally, we expected the drug should arrive at the chamber in 50 min at a flow rate of 0.1 μL/sec, which was confirmed by injecting a dye solution to our perfusion system. However, TRIP arrived later than 70 min.”

  1. On Fig. 2, is misleading to include in the same figure, the viability graph, and the microscopic images, since the viability assay was not conducted along MEIRS experiments. Authors did not measure viability of cells at T=0 and T=1000 min of the MEIRS experiments.

Action: We moved all the microscopic images to Figure 4 (e).

  1. Just a comment: Would be interestingly to have run more replicated MEIRS experiments, to enable to stop some experiments at specific time intervals, to analyse the cells morphology, cells number and cells viability.

This is a good point. However, it’s time consuming to measure this with the current setup. We have plans to work on the experiment with a multiwell-integrated metasurface setup in the future.

Action: Leave to the future work.

  1. Line 363, be careful with interpretation since, at the beginning of the experiment, cells were already full confluent. When this happens, usually cells start to dye and not to grow. Furthermore, the cells division usually takes 20 hr.

We have used A431 cells for our previous experiments as well, and usually we did the measurement with a confluent coverage of cells. In the absence of any drug, we have never observed cell death during our experiment. In cell culture flask, A431 cells can be kept at confluent coverage over many days without dying. What appears to occur is that the cells enter G0 quiescent phase; the cells stop proliferating, but they do not die either.

  1. To help interpreting, it is important to know if cells were putted upside down just before starting the MEIRS experiment? This is relevant for data interpretation since cells always need some adaptation period when putted in a new environment.

After setting up the flow system, we waited 1h so that cells have time to adapt to the new environment.

Action: Texts are added. “The cells in the chamber were placed upside down and sat for an hour to adapt to the new environment before any measurement.”

  1. Also, protein denaturation leads to increase absorbance of amide bands. Consequently, absorbance increase could be from this phenomenon and not necessarily from protein new synthesis. For example, reference 33, indicates that TRIP starts by inducing unfolded protein that eventually leads to apoptosis. Therefore, the initial absorbance increase could be partly from protein unfolding, or even to increase protein expression, due to the cells response to the stress (TRIP).

This is an interesting point. Protein misfolding and aggregation were observed in response to TRIP as early as 30min after treatment in our previous study (Ref. 33). Although rather speculative, the second peak in the protein signal may be related to such protein aggregates formation.

Action: We have added some speculation to the origin of this second protein peak: “In particular, the protein signal appears to have a second peak at around t = 140 min, which is not observed in lipid and refractive index signal. The origin of this peak is not clear, but it may be related to the accumulation of misfolded proteins and protein aggregates induced by ER stress due to TRIP [33].”

  1. Considering the hypotheses of “MOA”, authors observed “peaks” at defined periods, that are associated to cells macromolecular rearrangements/ or new synthesis. Authors did not prove with the present work that these rearrangements were due to Ca2+ or due to GPCR, or even at the process beginning (when absorbance increase) if were due to protein denaturation or protein synthesis. So, I do not think that MOA was minimally proved. Consequently, I also do not agree with the manuscript title.

We agree with the reviewer.

Action: The title has been changed to: ”Probing the Drug Dynamics of Chemotherapeutics Using Metasurface-Enhanced Infrared Reflection Spectroscopy of Live Cells.”

  1. Line 390, please add a reference concerning TRIP inducing autophagy.

Ref 33 mentioned autophagy induced by TRIP.

Action: Texts are added. “ The second phase’s origin is likely from the cell shrinkage or partial detachment. From the previous studies, ER stress and UPR caused by TRIP induce autophagy and apoptosis [33 , 34 ], which can lead to cell shrinkage and detachment.”

  1. In introduction section, authors points in line 64 “basal membrane”, but I wonder if in this type of cultured cells there are a basal membrane. If so, I would request authors to include a reference.

This is our poor choice of word.

Action: Replace “Basal membrane” with “cell membrane on the basal side ”.

  1. All the Discussion section is a lithe confusing. Authors should review it.

Action: Discussion 4.1 has been rewritten. Please check the marked PDF.

Despite of all these comments, I think that the work is valuable. However, before publication all these doubts should be clarified, and the text corrected according to it.

Round 2

Reviewer 1 Report

From my point of view, this revision is okay now and can be published.

Author Response

We thank the reviewer for the comment!

Reviewer 2 Report

The revised version of the manuscript is an improved version. 

Now, I have only minor comments.

For example, in the introduction section, I suggest adding extra references focusing MOA prediction by IR spectroscopy, as e.g. pointed in http://hdl.handle.net/10.400.21/12742  .

The difference between TRIP signal in experiments with and without the cells, could most probably due to TRIP intake by the cells. This is a very interestingly observation, that could enable to observe the drugs passage from the extracellular medium to intracellular.

I think that the conclusion section is to much simplified. It is wrongly only focusing the potential advantages over the MTT assay.  I also do not think that the assay is an alternative to MTT. But instead, a complementary and very appealing tool. I would suggest, authors to include in the discussion section, the necessity for future research, by evaluating along the experiment, the cells viability (by MTT or another assay).

Author Response

  1. For example, in the introduction section, I suggest adding extra references focusing MOA prediction by IR spectroscopy, as e.g. pointed in http://hdl.handle.net/10.400.21/12742 .

We thank the reviewer for the comment.

Action: The reference is added to the introduction.

“IR spectroscopy has also been applied to the investigation of the actions of various stimuli on cells and the resulting cellular responses, such as detecting immune cell activation, bacterial response to various antibiotics, and the modes of action (MoAs) of different families of drugs to the targeted cell lines [13-17].”

The difference between TRIP signal in experiments with and without the cells, could most probably due to TRIP intake by the cells. This is a very interestingly observation, that could enable to observe the drugs passage from the extracellular medium to intracellular.

  1. I think that the conclusion section is to much simplified. It is wrongly only focusing the potential advantages over the MTT assay. I also do not think that the assay is an alternative to MTT. But instead, a complementary and very appealing tool. I would suggest, authors to include in the discussion section, the necessity for future research, by evaluating along the experiment, the cells viability (by MTT or another assay).

We agree to the comment.

Action:

The below is added to the end of Discussion section: “In future studies, it will be interesting to compare MEIRS with other conventional reporter-dye-based cellular assays in monitoring cell viability and different modes of cell death.”

The conclusion is changed to: “MEIRS provides an appealing complementary tool to MTT assay for measuring the course of cytotoxicity in real time.”

This manuscript is a resubmission of an earlier submission. The following is a list of the peer review reports and author responses from that submission.

Round 1

Reviewer 1 Report

Reviewer Comments

The subject of the paper is of substantial importance, since cell chemistry monitoring allows the development also of new anti-cancer pharmaceuticals (cytostatic drugs). The MEIRS spectra show the cellular chemistry as dominated by protein and lipid signatures. I am surprised not to see any water absorption bands (probably compensated by recording a background spectrum with the culture medium only, which should result in an overcompensation of water bands, especially to be expected for the amide I band). Another point is the recording of the CRR spectra along dissociating the cells with trypsin. I suppose that the trypsin signatures have been separated in a difficult manner from the cellular protein spectra – please comment on the magnitude of these spectral signatures. The IR-tag subgroups can be differentiated by three  C≡O stretching frequencies, which are relatively invariant and span a narrow range of 2033–2036, 1957–1967, and 1926–1941 cm–1, while the C≡N stretching wavenumber can be found above 2150 cm-1 that is overlapped with the plasmonic resonance region, which should be discussed in more detail in combination with the reference spectrum given in Fig. 3. The authors should provide an estimate on the sample thickness, which is probed by the reflection mode (how much of integral cells have been monitored or cellular membrane?  This should be rated against the cell size of about 20 µm). Overall, the authors could show changes in cellular chemistry over the time period of about 7 hours with focus on (whole cellular?) protein and lipid content after treatment with two different cytostatic drugs.

In addition, checking the supplemental diagrams, I found that the spectral traces for the repeat experiments were not too convincing. 

Reviewer 2 Report

Please see the attached pdf document. 

Reviewer 3 Report

Dear authors,

Although the development of the MEIRS surface and its application to living cells is an innovative approach there is a number of problems with the presented work.

The main problems are

  • The use of overcomplicated data analysis method based on an obscure use of PCA instead of peak area. Also, more spectral data should be shown.
  • The data interpretation is too far-fetched. For example, there is no demonstration that changes in cell refractive index are connected to intracellular cell signaling. Either you provide definitive evidence of such connection or use more cautious language, and give alternative, non-ad-hoc explanations. Cell movements, detachments might be more likely explanations to the observed changes.
  • There is a fundamental problem in comparing cisplatin at concentration 1/26th its IC50 and TRIP at half its IC50. From fig 1, it can be seen that 2.5 µM of cisplatin is not decreasing cell viability while TRIP at 5µm is causing a 30% cell viability decrease. These two conditions can therefore not be compared, and no conclusion can be drawn on the capability of MEIRS to elucidate differences in the drug mechanisms of action. Cells show basically no reaction to cisplatin in your dynamic data (and in fig 9). The experiment needs to be redone with a cisplatin concentration that has a similar effect on cells than that of TRIP.

The list of reference 2-7 should be updated with modern references. For example, the paper of Rigas et al. which is clearly ridden with errors should not be cited in a serious analysis in 2021.

Reference 12 should be edited (12. Lehmkuhl, B.; Noblitt, S.D.; Krummel, A.T.; Henry, C.S. Fabrication of IR-Transparent Microfluidic Devices by Anisotropic Etching of Channels in CaF2. Lab Chip 2015, 15, 4364–4368. doi:10.1039/c5lc00759c. Version September 26, 2021 submitted to Cells 16 of 17)

As indicated by recent real-time measurements of live cells, low signal-to-noise ratio 39 (SNR) of ATR-FTIR necessitates a multi-bounce ATR element [13].      
            Alternatively, brighter light source (IR supercontinuum laser or QCL, synchrotron) could be used. Coupled to a microscope it could be integrated in a high-throughput set-up. MicroATR spectroscopy has been used multiple times to probe living cells down to single bacterial cell level1 without the need for IR amplifying meta-surfaces. Given the complex data analysis, limited penetration and other limits of the proposed technique, a better discussion of the advantage of using MEIRS should be given.

  1. Meneghel J, Passot S, Jamme F, et al. FTIR micro-spectroscopy using synchrotron-based and thermal source-based radiation for probing live bacteria. Anal Bioanal Chem 2020 41226. 2020;412(26):7049-7061. doi:10.1007/S00216-020-02835-X

What is the difference of MEIRS and SEIRA?

Since MEIRS requires measurements in reflection, wouldn’t the double optical path through water be detrimental to SNR and cause saturation?

TRIP’s toxicity to 66 cancer cells is affected through the induction of endoplasmic reticulum (ER) stress

Ref 24 is a general reference about ER stress, not a description of the mechanism of action of TRIP. Is there such a review? Ref 22 would maybe be better?

Signal in the 1900-2100 cm-1 from the TRIP?

Line 123-126 “An FTIR spectrometer (VERTEX 70, Bruker) coupled with an IR microscope (HYPERION 3000, Bruker) with a liquid-nitrogen cooled mercury-cadmium-telluride (MCT)
detector were used to collect the IR spectra with Dw = 4 cm-1 spectral and Dt = 1 min
temporal resolutions”

            What microscope aperture was used? How many cells were found in the field of view of the projected aperture and measured in one spectrum? Were cells static during the measurements? Did cell enter or leave the FoV, detached or multiplied?

Line 141-149 “Time-dependent IR absorbance and its second spectral derivative inside each spectral window (proteins: 1498 < w < 1807 cm−1, TRIP intrinsic absorption ("IR-tag"): 1845 < w < 2135 cm−1, plasmonic resonance: 1845 < w < 2231 cm−1, and lipid: 2756 < w < 3064 cm−1) were obtained by the linear regression with a set of corresponding cell response reference (CRR) spectra Lwin(w), where the variable win labels the four above-listed spectral windows. Specifically, the time-dependent absorbance A(w, t) or its second derivative ¶2A(w, t)/¶w2 spectra were projected onto their corresponding CRR spectra Lwin and ¶2Lwin/¶w2, respectively, and the time-dependent scores Swin(t) for each drug were obtained for each spectral window”

            Why use this convoluted method? Why not use directly the area of the peaks? What does ‘projected onto their corresponding CRR spectra” mean? What algorithm was used for the projection, what software, what validation method? Is it a projection in the PCA score space? Are the CRR the principal components of a PCA? PCAs generally have positive and negative loadings. Was a PCA mae without normalizing the data leading to all positive loadings?

Maybe the paragraph from lines 153-163 should come before the paragraph on lines 141-149.

Line 160 “Such reference spectra, obtained from the cell dissociation with trypsin, correspond to 161 the spectral difference between reflectances from metasurfaces with and without cells.”

            Then why use such a complex procedure? Why not use peak areas directly?

Line 164 “The second spectral derivative spectra were used for the protein and lipid windows to avoid the contribution from the shift in the underlying plasmonic resonance.”

            What shift? Is that a baseline drift or a peak position shift? A baseline drift can be corrected by baseline subtraction either using linear or more complex polynomial baselines. I’m sure the authors could simulate the Fano resonance lineshape and use it for subtracting a baseline. Even a linear baseline would probably work ok and allow interpreting spectral data that actually look like spectral data. Was this simple data analysis tested? A validation using this simpler approach could be useful. Using the method described by the authors do we know what we are looking at? Was their method validated by some external control or just because it looked right to the authors? The validation of the approach by comparing with baseline corrected peak areas should be presented in Supplementary data

The selected dose for cis is 2.5 µM, even smaller than for TRIP (5 µM) while TRIP seems more cytotoxic. The selected cisplatin concentration seems to have no effect on cell viability while the selected concentration for TRIP seems to have a 35% decrease in cell viability. A similar cell viability decrease would be reached at ~50 µM for cisplatin. Why this choice? This may make the effect off cis undetectable.

Line 199 “Differential absorbance spectra A(w, t) = − log[R(w, t)/R(w, t = 0)]”

            Why differential absorbance? These are log(reflectance) spectra.

Line 202-204 “Therefore, to assist the interpretation of the collected data, we have chosen to extract a series of temporal evolution curves Swin(t) associated with phenotypic cellular changes.”

            This method doesn’t seem to be fully validated unless there is a reference. The evolution of peak areas should be presented in Supplementary data.

Line 209-211 “The extracted scores Swin(t) are presented in Figures 3, 4, 6, and 7 for the bio-orthogonal (IR-label) spectral window, and the three bio-relevant (proteins, lipid, plasmonic resonance shift) windows.”

            The use of second derivative data makes interpretation difficult. Please present baseline corrected spectra in supplementary data in order to validate the results for such a new technique, by allowing comparison with known IR spectra.

Why water is not detected at all in this set-up? Was the background recorded on a dry substrate or hydrated substrate? Water has a band at 2100 cm-1, can it play a role in the measurement?

In figure 3, please present the signal of the TRIP alone, without cells, as presented in supplementary figures.

Line 236 “Presently we do not have a satisfactory explanation for the rapid initial increase of the SIR−tag(t) signal in TRIP-treated cells.”

            From figure S4 it takes 30 to 60 min to the TRIP to penetrate in cells and reach the plateau phase after the TRIP solution reached its maximum at 80 min. Internalization of the drug could take up to 60 minutes. This kinetic might be take the shape of a Hill active diffusion mode rather than a Fick passive diffusion mode. This was described in Clède et al. Applied Spectroscopy 2019 for another rhenium carbonyl compound. This may imply that the TRIP is internalized by the cells through protein channels. This could be verified by fitting the data with an active and a passive diffusion model.

Line 254-256 “Such changes include: increase or decrease of the cells’
adhesion to the metasurface, cytoskeletal rearrangements, and protein translocation
to/from the cellular membrane.”

            Changes in temperature, number and size of cells in the FoV could also affect reflectivity and refractive index. Was the number of cells in the FoV during the 8 hours of the experiment monitored? What is the doubling time of the population in these conditions?

Line 262 “MEIRS provides information that is very similar to that obtained from DMR-RTCAs”

            Do the authors have a reference to prove that?

Line 265 “inclusing surface-plasmon resonance (SPR)”

            Typo, “including”

Line 283: “This observation indicates that TRIP is a much more potent drug that cisplatin, most likely with a very different MoA.”

            This is plain wrong, this observation just reflects that you the cells were treated with a cisplatin dose that was much lower than IC50 than for the TRIP dose. The 2.5 µM cisplatin concentration caused no reduction in cell viability while the 5 µM TRIP concentration caused a 30% loss in viability. The authors need to redo the experiment with a cisplatin concentration that causes the same level of cytotoxicity if they want to be in position to draw such conclusions.

Line 284 “This conclusion about relative potency of the two
drugs is consistent with their respective IC50 values: ICcis 50 -Pt = 65.4 ± 2.5 µM versus
ICTRIP 50 = 10.8 ± 2.8 µM.”

            This raises the question whether TRIP is too cytotoxic for clinical use. Cisplatin can be used since it strikes the right balance between cytotoxicity toward cancerous cells and normal cells. Although a IC50 of 10µM is not that strong. Again, treating the cells with a cisplatin concentration smaller than that of TRIP while cisplatin IC50was lower than that of TRIP doesn’t allow to compare the potency  of the two drugs and draw firm conclusions .

Line 307 “This, in combination with a rapid early spike of Spl(t) related to intracellular signaling, indicates that the MoA of TRIP complex is qualitatively different from that of
cisplatin.”

            How do you know that the spike is related to intracellular signaling?

Line 309 “At the same time, the effect of cisplatin on the cells is statistically distinguishable from that of the blank L-15 medium even at t = tfinal (i.e. Dt = tfinal - t0 455
min after the arrival of the drug) despite the fact that no morphological changes can be
observed using phase-contrast microscopy this early into the treatment.”

            The data presented in figure 4 and S1 do not show a statistically significant difference in control and cisplatin treated samples. Differences in focus, could easily explain such apparent refractive index changes.

Figure 6. and line 320 “The increase (decrease) of the Sprot(t) score signal corresponds to the increase
(decrease) in protein IR absorption.”

Again,  the use of second derivative and signal extracted from PCA is confusing. Second derivative peaks are negative. The CRR signal presented in fig 6 is positive. Was the signal multiplied by -1 at any point or should we understand that an increase in the signal strength means a decrease in protein signal?

Line 328 “Because the latter was most likely
related to intra-cellular signaling, we conclude that the spike in Sprot(t) has the same
origin.”

            The authors never conclusively prove that he change in refractive index meant changes in cell intracellular signaling. I could be related to changes in a number of protein-surface interactions, cytoskeletal changes, relocation of proteins (gravity?), cell movements, cell shape changes … What is the probing depth probed of system?

Line 337 “Time-dependent lipid absorption data does not distinguish between control and

cisplatin-treated groups, but distinguishes the TRIP-treated group from the other two.”

There is no evidence to suggest that this is not an effect of the too low cisplatin concentration used.

Figure 7.

                Please label the peak position in the insert.

Line 350. ”We observe from Figure 7 that only the TRIP-treated cells, but not the control or cisplatin-treated ones showed a significant decrease in the lipid signals.”

A result of the low cisplatin concentration.

Line 362. “In this work, we have demonstrated that MEIRS technique can be used to characterize the dynamics of drug arrival to the cell, as well as its effect on intracellular signalling and gross morphological changes, such as the evolution of the cell adhesion, shape, and cytoskeletal architecture.”

            Signaling, not signalling

            There is no definitive evidence that MEIRS signal is sensitive to cell signaling in this paper, only conjectures. The authors should be more cautious. Line 290 the authors say “As we discuss in Section 4.1, the most likely origin of the spike is intra-cellular signalling” and in section 4.1 they say “In this work, we have demonstrated that MEIRS technique can be used to characterize the dynamics of drug arrival to the cell, as well as its effect on intracellular signalling”. There is nowhere near a demonstration of the link between changes in the signal and internal cell signaling at this point in the paper or later.

Most of the differences between the cisplatin and TRIP reactions could be ascribed to the fact that the authors sued a 2.5 µM cisplatin concentration while cisplatin has a 65µM IC50, and a 5 µm TRIP concentration while TRIP IC50 is ~10 µm. Thus TRIP was used at half its IC50 and cisplatin at 1/26th its IC50. The authors can therefore not assign the differences in reaction to the mode of action of the drugs but rather to the extremely different concentrations used. The data also reflect that since cisplatin treated cells basically show no reaction to the treatment. Baseline drifts and various imprecisions in measurement could explain the non-specific signals observed in figures 3 to 7. The authors need to show spectral data if they want to claim observing changes in the cells. The kinetic signals extracted after an over-convoluted and complex data analysis are not convincing.

Round 2

Reviewer 1 Report

For the revised manuscript, I cannot see substantial improvements and still considerable deficits in scientific “craftsmanship”. A week point is still the too low concentration of the cis-platin compound so that such this particular experiment is not really contributing to create knowledge compared to the control studies.

One point is also the introduction, which does not provide a description of the state of the art. There are significant publications dealing with spectral cytopathology, e.g., reaction to virus infection or bacterial responses to antibiotics, new developments in vibrational histopathology, immune cell activation to mention a few, where infrared spectroscopy has been successfully applied for live cell monitoring. In addition, that bulky ATR elements would be required for such investigations is not true and single-bounce crystals have been used recently. Multi-module EC-QCLs have also been introduced to cover a spectral interval of 1000 cm-1 especially around the spectral fingerprint region (see also applications for spectral histopathology with latest instrumental developments). Interestingly, it is claimed that MEIRS allows smaller field penetration of roughly 100 nm only, and I would be happy to see an experimental proof.

Further problems with presentation of results: The dimension of a pixel of the metasurface arrays cannot be 300 µm x 300 µm (presumably these were nm as dimension), but this shows also the deficits of a not too well prepared manuscript. For Fig. 1 (d) I cannot see any water features. The authors explain that the latter spectrum was from measurements with cells on the metasurface versus the reflectance spectrum with no cells. For a reflectance measurement two single beam measurements are required, but the authors do not explain what background measurement had been recorded. Was the metasurface dry without medium-coverage? The reflectance spectrum in subplot (c) shows some substantial water absorption at 1640 cm-1 and spectrum (d) seems to have the water absorption compensated, for which I would expect a much smaller amide I band due to overcompensation. If the background spectrum had been the same for the measurement with adherent cells and without, this could certainly be reduced to the –log (ratio of two single beam spectra).

Well, I was also checking the reference (18) with previous work from the authors (see Fig. 2 there) where the description of the calculation of the absorbance spectrum is quite differently explained compared to the insufficient way, which actually makes me disappointed to see the incomplete explanation given in the revised manuscript apart from the duplication of diagrams, which is a case of self-plagiarism.

I am also questioning the preparation of the CRR (cell response reference) spectra obtained from different procedures, for example, from a PCA of spectra “produced by dissociating the cells with low-concentration trypsin”. Here it is to be expected that spectra after dissociating the cells with dominating water bands will show up, which certainly will affect the first PC component for distorting band intensities around amide I and II bands. I am surprised that not cell spectrum as shown with Fig. 1 (d) could be used for further data evaluation of the spectra obtained during cell monitoring.  “The time dependent IR absorbance signal inside each window were obtained by least-square linear regression from a set of corresponding CRR spectra”, which must be interpreted in a way that just scaling factors had been calculated for the course of monitoring.  A projection operation is different from the least-square fits mentioned lines above in the manuscript. The authors show only one score plot for the different spectral windows, but when inspecting the supplemental diagrams, the quality concerning Signal-to-noise and repeatability is much too poor to be accepted for publication and several more repeat experiments are required for a high-quality presentation of cellular responses. Fig. 3 makes me think what is wrong with the time –dependent signal features, as drug arrival time is after 45 min but response within the time-dependent stepwise increase for the TRIP agent without cells is after 80 min, with a further 10 min delay for the PRIP experiment with cells providing a double concentration (by cellular uptake and enrichment, which does not make sense to me) . For the complete coverage of adherent cells with a thickness of about 20 µm, I find no explanation of such time dependence; the ascending part should also provide an estimate of the increasing concentrations of TRIP at least within the claimed monitored spectral penetration depth. This could also provide some insight into the processes, which may not be from intracellular uptake but from leakages between cells. By the way, time signal dependencies differ very much if Fig. 3 and Fig. S8 are compared, for the latter also no repeat measurements are shown for allowing a judgement of repeatability. S8 has a header “TRIP 231 cm-1 integration” which creates the impression that the carbonyl bands have been used for signal generation, however in Fig. 4 the sign-inversed features are displayed with the plasmonic cell response reference spectrum with band maximum clearly at 2050 cm-1! Such discrepancies cannot be accepted. Explanations given on page 8 for explaining the phenomena are rather speculative. A way to resolve this problem could be to isolate the washed cells and to analyze the intracellular TRIP concentration by mass spectrometry. There are inconsistencies with the ordinate labelling in the manuscript and the supplemental diagrams, which must be corrected and brought in line.

When inspecting the change in protein signals, the features seem to show a clear distinction between the three categories of experiments, but looking critically at Figure S2, this is not the case and is much in doubt and not unique enough to allow the support of speculation of the authors. Similar, such situation can also be found for the lipid signal (see Fig. S3). Such weak evidence is not acceptable and must be supported by higher quality data based also on repeated experiments. Morphology changes certainly provide an indication of cellular response to the anti-cancer agent, but it is also evident that the dense cellular carpet from the experiment beginning has been dissolved making it difficult to separate intra- and extracellular processes during monitoring. Therefore, a much more detailed and critical discussion must be given including better spectral data to get this manuscript published.